# A Novel LSTM-Based Machine Learning Model for Predicting the Activity of Food Protein-Derived Antihypertensive Peptides

**DOI:** 10.3390/molecules28134901

**Published:** 2023-06-21

**Authors:** Wang Liao, Siyuan Yan, Xinyi Cao, Hui Xia, Shaokang Wang, Guiju Sun, Kaida Cai

**Affiliations:** 1Key Laboratory of Environmental Medicine and Engineering of Ministry of Education, School of Public Health, Southeast University, Nanjing 210009, China; wangliao@seu.edu.cn (W.L.); 213203600@seu.edu.cn (S.Y.); 220214004@seu.edu.cn (X.C.); huixia@seu.edu.cn (H.X.); shaokangwang@seu.edu.cn (S.W.); gjsun@seu.edu.cn (G.S.); 2Department of Nutrition and Food Hygiene, School of Public Health, Southeast University, Nanjing 210009, China; 3Department of Epidemiology & Biostatistics, School of Public Health, Southeast University, Nanjing 210009, China; 4Department of Statistics and Actuarial Sciences, School of Mathematics, Southeast University, Nanjing 210009, China

**Keywords:** antihypertensive peptides, structure–activity relationship, machine learning, LSTM algorithm

## Abstract

Food protein-derived antihypertensive peptides are a representative type of bioactive peptides. Several models based on partial least squares regression have been constructed to delineate the relationship between the structure and activity of the peptides. Machine-learning-based models have been applied in broad areas, which also indicates their potential to be incorporated into the field of bioactive peptides. In this study, a long short-term memory (LSTM) algorithm-based deep learning model was constructed, which could predict the IC_50_ value of the peptide in inhibiting ACE activity. In addition to the test dataset, the model was also validated using randomly synthesized peptides. The LSTM-based model constructed in this study provides an efficient and simplified method for screening antihypertensive peptides from food proteins.

## 1. Introduction

Globally, hypertension has been ranked as one of the major chronic diseases. It has been estimated that about 1.4 billion adults are suffering from hypertension worldwide and the prevalence is still on an upward trend [1]. The renin–angiotensin system (RAS) plays a major role in the regulation of blood pressure. Angiotensin II (Ang II), which is a potent vasoconstrictor in the RAS, is generated from Ang I with the action of angiotensin converting enzyme (ACE) [2]. Clinically, the inhibition of ACE activity to suppress the formation of Ang II has been considered an efficient strategy for the management of high blood pressure. Thus, synthetic ACE inhibitors have been used as a first-line pharmaceutical drug for hypertension therapy [3].

Notably, peptides that could inhibit ACE activity were identified from snake venom in 1971 and were characterized as ACE inhibitory peptides [4]. Since then, a large number of ACE inhibitory peptides have been identified from various natural protein sources, including food proteins such as milk proteins, egg proteins and soy proteins [5]. Compared with synthetic drugs, food protein-derived ACE inhibitory peptides are considered to have fewer side-effects and lower production costs, which makes these peptides a promising alternative for antihypertensive drugs.

As a representative category of food protein-derived bioactive peptides, research on ACE inhibitory peptides is diverse and mainly focuses on peptide identification, mechanistic study and clinical trials [6]. Particularly over the past two decades, enormous efforts have been paid to delineate the relationship between the structure and activity of ACE inhibitory peptides. Since it has been widely accepted that the biological activity of a chemical structure can be described by its chemical features, such as its composition, electronic attributes and hydrophobicity [7], the value that inhibits 50% of the ACE activity (known as the IC_50_ value) has been used as an output that correlates with the structural features of the peptides. Based on this principle, quantitative structure and activity relationship (QSAR) modelling was applied in order to predict the IC_50_ value of the ACE inhibitory peptides, and several models have been established [8,9]. However, representing the structural features of a peptide is a complicated process. In addition, the use of different strategies for peptide representation may result in variations in the accuracy of these models.

Artificial neural networks (ANN) are algorithmic mathematical models that mimic the behavioral characteristics of animal neural networks and perform distributed parallel information processing. ANN relies on the complexity of the system and achieves the purpose of processing information by adjusting the interconnected relationships between a large number of internal nodes. The deep learning-based ANN has been widely applied in the field of biomedicine. Several deep learning-based models have been constructed to predict the activity of antioxidant peptides [10,11], anticancer peptides [12] and antibacterial peptides [13]. The long short-term memory (LSTM) network is a special type of recurrent neural network (RNN) that is capable of learning order dependence in sequence prediction problems. Compared with shallow learning, LSTM has a deep learning framework with a large number of hidden layers, allowing it to learn more complex non-linear patterns [14]. Notably, the LSTM-based model has been constructed for the discovery of antimicrobial peptides [15], suggesting the feasibility of applying LSTM in predicting the activity of bioactive peptides.

Collectively, an LSTM-based prediction model was constructed in the present study, which could provide an efficient and simplified structure and activity model for ACE inhibitory peptides, as well as enabling further exploration of the application of LSTM networks in the field of bioactive peptides.

## 2. Results

### 2.1. An Overview of the Dataset

In total, 3429 peptide sequences with their corresponding IC_50_ ACE inhibitory values were retrieved from the database and used in this study. As shown in Figure 1A, the IC_50_ values of the peptides were variable and ranged from less than 1 μM to above 1000 μM. However, the IC_50_ values of most of the peptides were less than 100 μM, indicating that these peptides have potent ACE inhibitory activity. In total, 2327 peptides in the data set were functional ACE inhibitory peptides.

The amino acid distribution of the peptides from benchmark datasets was also analyzed. It is obvious that proline appeared most frequently, accounting for 19.2% of all the amino acids, which is strikingly higher than the frequency of the other amino acids (Figure 1B). This finding is in line with previous reports that proline appears to be a frequent amino acid present in various bioactive peptides [10,16]. On the contrary, methionine is absent in the dataset, and the underlying reasons for this are yet to be determined (Figure 1B).

### 2.2. Performance Evaluation of the Model

The variations in train loss and test loss for the LSTM model show that as the training cycle progresses, the variations in train loss and test loss decrease (Figure 2), which indicates that the prediction accuracy of the LSTM model could be improved through training. However, the curves of the train set and test set were not superimposable, which might be due to the limited number of data included in the test set.

The performance of the model was then evaluated by five-fold cross-validation. The mean accuracy, average sensitivity and average specificity of the model was 85.20%, 84.92% and 85.43%, respectively. Furthermore, the RMSE was 0.18.

In addition, for the 343 peptides included in the test set, the ratio of the predicted IC_50_ and the reported IC_50_ was plotted. As shown in Figure 3, the ratio of 256 peptides were distributed within the range of 0.75 and 1.25, which suggested the accuracy of the model.

### 2.3. Model Validations

Based on the literature search, 54 peptides were retrieved that were reported with both their in vitro ACE inhibitory IC_50_ value and their in vivo blood-pressure-lowering effect. We then applied our LSTM-based model to predict the IC_50_ value of these peptides. As shown in Table 1, the ratio of the predicted IC_50_ and the reported IC_50_ of 38 peptides were distributed between 0.80 μM and 1.20 μM, among which, the ratio of 19 peptides were between 0.90 and 1.10. These results indicate the potential of our LSTM-based model to predict the IC_50_ value of antihypertensive peptides with in vivo activity.

Finally, 20 peptides were randomly generated and synthesized. The LSTM-based model was then applied to predict the IC_50_ value of these peptides. The experimental IC_50_ value of each peptide was provided via the HPLC-based assay. As shown in Table 2, the ratio of the predicted IC_50_ and the experimental IC_50_ of 15 peptides were between 0.75 and 2. Such a result suggests the feasibility of predicting the ACE inhibitory value of a random sequence using the model developed in the present study.

## 3. Discussion

Food protein-derived antihypertensive peptides are one of the representative categories of bioactive peptides. Research on antihypertensive peptides has been ongoing for about five decades. Research into the structure and activity relationship of peptides has long been a prominent research area. The QSAR modelling of food protein-derived antihypertensive peptides started about two decades ago. Initially, research concentrated on the structural features of di- and tri-peptides using partial least squares regression [39]. However, the efficiency of the model was too limited to be used for high throughput prediction. Notably, machine-learning-based techniques have been applied widely across multiple areas. Importantly, several machine-learning-based models have been constructed that could be used to predict the activity of antioxidant, anticancer and antimicrobial peptides [11,13,15], which indicates the feasibility of applying machine learning algorithms in the QSAR modelling of bioactive peptides.

It has been previously reported that a machine learning model based on the support vector machine algorithm was developed to predict the antihypertensive activity of food protein-derived bioactive peptides. However, the accuracy of the model was less than 80% [40]. In a later study, the extremely randomized tree algorithm was applied, and the performance of the model was improved to 85.0%. However, this model consists of 51 feature descriptors, which makes the model complicated [41]. Notably, we utilized an LSTM deep learning model to investigate the relationship between peptide structure and bioactivity in the present study. As a special type of RNN, LSTM has the advantage of capturing historical information from prior inputs, allowing it to influence the current input and output applications for speech recognition, natural language processing and time series prediction [42]. In real-life data analyses, when the time interval is long due to the gradient vanishing problem, RNN does not have the ability to memorize the previous information well. To overcome this disadvantage, LSTM was proposed by combining short-term memory with long-term memory through gate control [43]. Importantly, our results demonstrate that the LSTM model achieved a correlation coefficient of 0.85 on the validation dataset. In addition, the LSTM model’s superiority over other models may stem from its ability to capture the sequential nature of peptide data, which allows it to detect subtle structural patterns that influence bioactivity. However, it is important to note that LSTM models have some potential drawbacks, including high computational costs due to their complex architecture and the possibility of overfitting if the dataset is not diverse enough. Therefore, future studies should explore ways to optimize LSTM model performance while controlling these factors.

Since the research on antihypertensive peptides originated from ACE inhibitory peptides, the database available for deep learning training was constructed based on the IC_50_ values of the peptides in the in vitro ACE inhibitory assay. Thus, despite the satisfactory performance of the model developed in this study, the biological significance of the model is yet to be determined. Furthermore, it is suggested that the current peptide database should be expanded by adding the results from biologically relevant assays, such as cellular experiments and animal studies, if available. The information from biologically relevant assays could be incorporated into the machine learning model in the future. On the other hand, studies in recent years have shown that there may be targets other than ACE for antihypertensive peptides in the context of reducing blood pressure [44]. Hence, it is also recommended that a database based on the other activity parameters of the peptides is constructed.

The LSTM-based model developed in this study also demonstrated high efficiency in predicting the IC_50_ values of randomly generated peptides. Therefore, this model could be potentially applied in peptide design, which may create novel opportunities for the screening of antihypertensive peptides. In addition, a recent study developed a machine learning empowered model capable of performing in silico gastrointestinal digestion of food proteins [45], which could be incorporated into our model to create a more comprehensive activity prediction system. However, only peptides composed of less than six amino acids were randomly generated, and the ability of the model to predict the activity of longer peptides is yet to be determined.

## 4. Materials and Methods

### 4.1. Benchmark Dataset

The peptide sequences used for data training in this study were obtained from a number of databases, including BIOPEP-UWM [46], FeptideDB [47] and BioPepDB [48]. In addition, we manually searched the literature to identify the peptides that were not included in the above databases. All of the peptides in the present study were manually curated, merged and cross-checked in order to construct a non-redundant data set. Furthermore, only peptides with an IC_50_ value less than 2000 μM were included in this study. Following data collection, the data was randomly divided into a training set and a validation set for the model in a ratio of 9:1.

### 4.2. Literature Searching Strategy

PubMed and Web of Science were searched in order to identify studies investigating the IC_50_ in in vitro ACE inhibitory assays, as well as the in vivo blood-pressure-lowering effect of food protein-derived bioactive peptides published up to April 2023. The search was performed using the following strings: “Bioactive peptides” AND “ACE inhibition” AND “Blood pressure reduction”. For model validations, peptides with known sequences that have been previously reported to exhibit in vitro ACE inhibitory IC_50_ values and significant in vivo blood pressure lowering effects were used in this study.

### 4.3. Representation of the Peptide Sequence

The 19 amino acids that appeared in all the peptides were mapped to different integers, as shown in Table 3. Then, each peptide sequence was converted into a digital sequence, which was then packaged into a Pytorch dataset, with a batch size of 32 as per the specified scale.

### 4.4. Machine Learning Algorithms

As shown in Figure 4, the LSTM network consisted of one input and output layer and a series of recurrently connected hidden layers. The hidden layers were memory blocks, with an input gate, an output gate, a forget gate and some self-recurrent memory cells. The input, output and forget gates provided read, write and reset operations for the memory cells, respectively. Figure 1 gives an example of an LSTM memory block with a single cell. There exists a recurrently self-connected linear unit-constant error carousel (CEC) at the core of each memory block. The outside interference was stopped by the self-recurrent memory cell and the status was held from one time point to another. This is why the LSTM can solve the vanishing gradient problem. Assuming that the model input at time *t* was *X*_*t*_ = (*X*_*t*1_, …, *X*_*tn*_)^⊤^, where *n* is the number of input dimensions, the input gate selected the information of input *X*_*t*_ to be saved into cell *C*_*t*_. The forget gate selectively forgot the state of the last moment cell *C*_*t*−1_. The forget gate learnt to reset memory blocks once their status was out of date. Furthermore, the forget gate prevented the cell status from growing boundless and saturating the squashing function. The components of the output *h**t* were controlled by the output gate; that is, the output gate controlled the ability of the cell state to influence other neurons.

To show the details, the training process of the LSTM model can be formulated with some equations. The input gate $i_{t}$ and the forget gate $f_{t}$ have the following formulas:*i*_*t*_ = *σ* (*W*_*i*_ [*h*_*t*−1,*X**t*_] + *b*_*i*_),(1)
*f*_*t*_ = *σ* (*W*_*f*_ [*h*_*t*−1,*X**t*_] + *b*_*f*_),(2)
where *h*_*t*−1_ is the output of the previous cell, *X*_*t*_ is the input and *b* and *W* denote the bias vectors and the weight matrices, respectively. Then, we can update the cell state *C*_*t*_ using the following formula:(3)$$C_{t}=f_{t}C_{t-1}+i_{t}\tanh\left(W_{c}[h_{t-1,X_{t}}]+b_{c}\right),$$
where *C*_*t*−1_ is the state of the previous cell, *b*_*c*_ and *W*_*c*_ denote the bias vector and weight matrix, respectively. Finally, the output gate *o*_*t*_ and output *h*_*t*_ can be defined as:*o*_*t*_ = *σ* (*W*_*o*_ [*h*_*t*−1,*X**t*_] + *b*_*o*_), (4)
*h*_*t*_ = *o*_*t*_tanh(*C*_*t*_),(5)
where *b*_*o*_ and *W*_*o*_ denote the bias vector and weight matrix, respectively. *δ*(⋅) and tanh(⋅) are the sigmoid and the tanh functions defined as follows:(6)$$\sigma \left(a\right)=\frac{1}{{1+e}^{-}},$$
(7)$$\tanh\left(a\right)=\frac{e^{a}-e^{-a}}{e^{a}+e^{-a}}.$$

The training frequency was set to 100 times. In each training session, the program disrupted the order of the entire database and reprocessed, encapsulated and allocated the training and validation sets in a 9:1 ratio. Then, the training set data was used to adjust the parameters of the model, and the validation set data was used to calculate the current error of the model. When the calculated error was less than the previous minimum error, the current model parameters and the output results of the model for the validation set were retained.

### 4.5. Model Evaluations

The model was evaluated in two dimensions. Firstly, the accuracy of the model in predicting the IC_50_ value of the specific peptide was assessed using the ratio of the predicted value and the reported or experimental IC_50_ value. The prediction was defined as “accurate” when the ratio matrix was within the range of 0.75 and 1.25, otherwise it was considered “inaccurate”. In this way, the regression task was converted to the classification task, which was further used for the five-fold cross-validation.

To assess the overall reliability of the model, a five-fold cross-validation was executed according to the literature, in which the original dataset was randomly separated into five equally sized sub-samples. Then, each sub-sample was used for the test data, whereas the remaining sub-samples were used for the training set. The cross-validation process was then repeated five times. The average of the five-fold cross-validation yielded the accuracy of the algorithm [49,50]. The results of the five-fold cross-validation were presented as the mean accuracy, average sensitivity and average specificity. In addition, the root mean square error (RMSE) of the model was calculated according to the following formula:(8)$$RMSE=\sqrt{\frac{1}{m}\sum_{i=1}^{m}{\left(y_{i}-{\hat{y}}_{i}\right)}^{2}},$$
where *m* is the sample size, *y* is the reported value and $\hat{y}$ is the predicted value.

### 4.6. The In Vitro ACE Inhibitory Assay

An online tool (https://www.genscript.com/sms2/random_protein.html accessed on 1 March 2023) was used to randomly generate peptides in order to test the efficiency of the model. The top peptides with a small number of IC_50_ values were selected for synthesis. In addition, since peptides composed of less than six amino acid residues possess stability in the gastrointestinal tract [51], the maximum length of the generated peptides consisted of five amino acids. The peptides used for validation were synthesized by Genescript with a purity > 97%. The ACE inhibitory assay was performed according to a previous study [52] with modifications. ACE, N-hippuryl-His-Leu tetrahydrate (HHL, Sigma-Aldrich, St. Louis, MI, USA) and the peptide samples were dissolved in 100 mM of boric acid containing 300 mM of NaCl (pH8.3). Firstly, 10 μL of the peptide solution was preincubated with 50 μL of 6.5 mM HHL at 37 °C for 5 min. Then, 5 μL of 0.1UN/mL ACE (preincubated at 37 °C) was added to the reaction system and incubated at 37 °C for another 30 min. The reaction was terminated by adding 85 μL of 1 M HCl. The concentration of hippuric acid (Hip, the reaction product) was measured by HPLC with a C_18_ column (5 µm, 250 mm × 4.6 mm). The sample (20 μL) was eluted by a gradient of solvent A (H_2_O with 0.05% TFA) and solvent B (acetonitrile with 0.05% TFA) at a flow rate of 1.2 mL/min. The absorbance at 228 nm was monitored. The concentration of HA was calculated based on its standard curve. The area under each peak was calculated, in which A = the area under the peak of the blank group (without the peptide) and B = the area under the peak of the peptide group. The ACE inhibitory ratio = (A − B)/A. The ACE inhibitory ratio of each peptide at different concentrations was measured. The IC_50_ value was defined as the peptide concentration inhibiting 50% of the ACE activity.

## 5. Conclusions

In this study, a novel model utilizing the LSTM-based deep learning network was constructed to predict the activity of food protein-derived antihypertensive peptides. The model achieved excellent performance in activity prediction, which was validated by both the test set of the benchmark dataset and the in vitro ACE inhibitory assay for randomly generated peptides. Therefore, this model could be used to screen antihypertensive peptides from various food proteins. In addition, this research provides a novel aspect for the QSAR study of antihypertensive peptides.

## Figures and Tables

**Figure 1 molecules-28-04901-f001:**
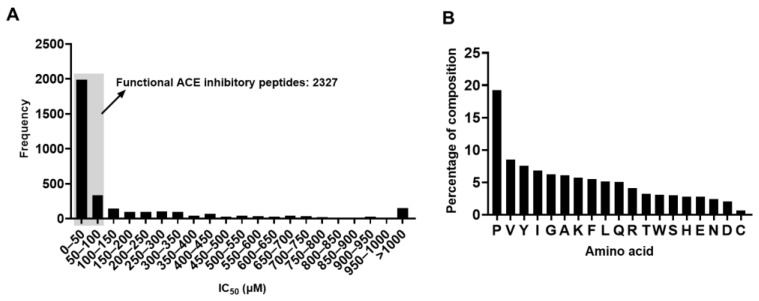
An overview of the dataset. (**A**) Frequency distribution of the IC_50_ values. (**B**) The percentage of each amino acid residue.

**Figure 2 molecules-28-04901-f002:**
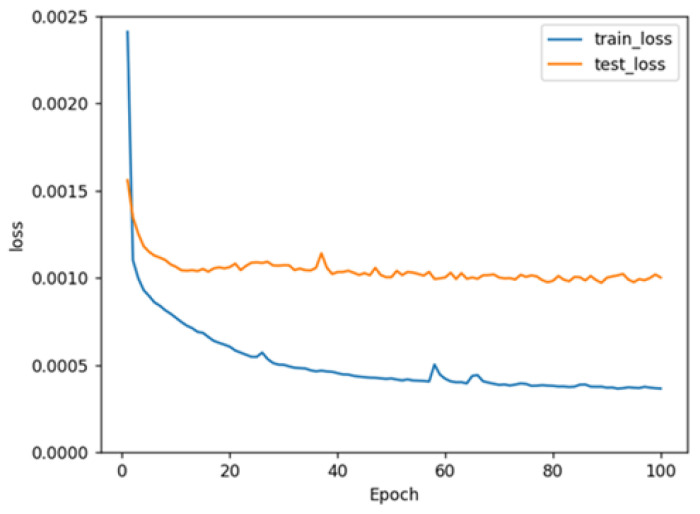
The training history showing train loss and test loss for the model.

**Figure 3 molecules-28-04901-f003:**
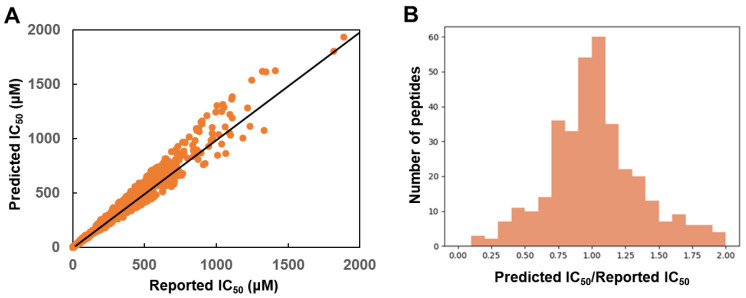
The evaluation of the test set. (**A**) The reported IC_50_ and the predicted IC_50_ of the test set. (**B**) The distribution of the ratio of predicted IC_50_ and reported IC_50_.

**Figure 4 molecules-28-04901-f004:**
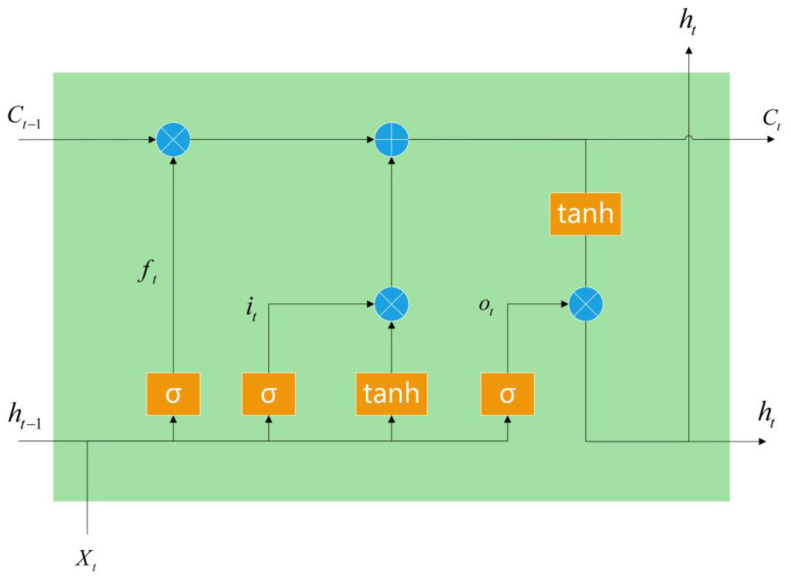
The network structure of the LSTM model.

**Table 1 molecules-28-04901-t001:** Peptides reported with in vitro ACE inhibitory activity and the in vivo blood-pressure-lowering activity.

Peptide Sequence	Predicted IC_50_ (μM)	Reported IC_50_ (μM)	Predicted IC_50_/ Reported IC_50_	Reference Reporting the IC_50_
KGYGGVSLPEW	0.23	0.70	0.33	[17]
LLVTLKK	0.42	0.95	0.44	[18]
LKY	0.36	0.78	0.46	[19]
PAGELHP	0.29	0.50	0.58	[20]
DAQSAPLRVY	7.60	12.20	0.62	[17]
RDGGYCC	0.56	0.84	0.67	[21]
WV	217.83	307.61	0.71	[22]
KF	20.08	28.30	0.71	[23]
LVY	1.30	1.80	0.72	[19]
IRW	0.44	0.61	0.72	[24]
FY	2.71	3.70	0.73	[23]
LEEFCC	1.36	1.85	0.73	[21]
GF	213.69	277.90	0.77	[25]
MLPAY	1.27	1.58	0.80	[19]
IQW	1.26	1.56	0.81	[26]
LRA	141.66	174.30	0.81	[27]
KIDKVVK	0.53	0.62	0.85	[18]
LKP	2.49	2.93	0.85	[26]
AFVGYVLP	12.62	14.41	0.88	[28]
LAK	42.46	48.00	0.88	[29]
NF	41.66	46.30	0.90	[25]
VY	10.24	11.30	0.91	[23]
HLNVVHGN	46.29	50.88	0.91	[30]
DKVGINYW	23.13	25.40	0.91	[17]
EKSYELP	16.54	18.02	0.92	[28]
PGSGCAGTDL	53.67	57.86	0.93	[30]
LSA	7.26	7.81	0.93	[19]
GAAELPCSADWW	10.25	10.95	0.94	[31]
KY	7.25	7.70	0.94	[23]
IVY	43.52	45.77	0.95	[32]
KW	10.28	10.80	0.95	[23]
VW	10.29	10.80	0.95	[23]
VDSDVVK	8.26	8.64	0.96	[33]
VF	42.00	43.70	0.96	[23]
LRLESF	5.21	5.39	0.97	[30]
YY	46.30	47.90	0.97	[27]
LDSPSEGRAPG	17.31	17.90	0.97	[20]
VIY	4.36	4.50	0.97	[19]
VELYP	5.23	5.22	1.00	[28]
WQVLPNAVPAK	1023.89	1010.00	1.01	[34]
TFQGGlPPHGIQVER	3.47	3.40	1.02	[29]
VISDEDGVTH	8.33	8.16	1.02	[35]
RLSGQTIEVTSEYLFRH	577.19	560.18	1.03	[36]
ILSKLK	4.28	4.02	1.07	[18]
AY	156.44	146.76	1.07	[37]
IISKIK	1.28	1.19	1.07	[18]
CTFSIPAQC	26.31	24.40	1.08	[38]
IY	2.96	2.70	1.09	[23]
LT	1.22	1.11	1.10	[22]
TVTNPARIA	16.33	14.50	1.13	[20]
LVLPGELAK	214.22	184.00	1.16	[29]
LQP	1.35	1.04	1.30	[19]
IPPAYTK	35.75	23.50	1.52	[29]
LVLPGE	20.79	13.50	1.54	[29]

**Table 2 molecules-28-04901-t002:** The predicted IC50 and the experimental IC50 of the randomly synthesized peptides.

Peptide Sequence	Predicted IC_50_ (μM)	Experimental IC_50_ (μM)	Predicted IC_50_ /Experimental IC_50_
LKPDQ	0.70	0.88	0.79
WD	0.63	0.51	1.23
GVPK	0.61	0.25	2.44
FI	0.61	0.31	1.95
PDFLI	0.60	0.33	1.83
HDHR	0.59	0.59	1.00
LKPNS	0.56	0.5	1.12
VYHEL	0.55	0.38	1.45
GPAY	0.54	0.37	1.45
LVL	0.51	0.32	1.59
LKL	0.49	0.56	0.88
FDKA	0.47	0.6	0.79
VAWKL	0.46	0.23	2.00
VHLAP	0.46	0.33	1.39
IQWCA	0.46	0.1	4.59
PLPLL	0.55	0.2	1.75
KLPAY	0.44	0.12	3.63
LKPI	0.43	0.39	1.11
FALPC	0.42	0.16	2.65
ALPD	0.72	1.55	0.46

**Table 3 molecules-28-04901-t003:** The digit representing each amino acid.

Amino Acid	Representing Digit	Amino Acid	Representing Digit
I	2	D	11
L	3	C	12
S	4	T	13
H	5	N	14
R	6	V	15
P	7	G	16
A	8	Q	17
W	9	K	18
F	10	Y	19
		E	20

## Data Availability

Data are contained within the article.

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
