# Peer review of "A Novel LSTM-Based Machine Learning Model for Predicting the Activity of Food Protein-Derived Antihypertensive Peptides"

_molecules, 2023, doi:10.3390/molecules28134901_

Round 1

Reviewer 1 Report

The work submitted by Kaida Cai et al. entitled A novel LSTM-based machine learning model for the activity prediction of food protein-derived antihypertensive peptides described the work with an LSTM and predicted potential peptide binders for ACE from the source of food. The result would be more impressive if authors could have developed a peptide inhibitor from a short de novo peptide library. Nonetheless, the LSTM model did not resemble experimental data well. Table 2 shows more than five potent peptides that do not agree with experimental data (ratio IC50 >2). Most importantly, the best binder found via experimental study is predicted as a weak binder through the LSTM model (entry 15 in Table 2). I don’t think this model is supreme for predicting a good peptide binder.

Few comments:

In many places, the IC50 values range has been given, but the unit is missed.

The images are blurry.

Authors would have considered the SAR studies to improve the binding ability and IC50 value of a selected best peptide binder.

The computational model needs to be updated and attained more specific algorithms for better prediction of peptide binder and close to experimentally validated value.    

The English are fine to read; it may need some small editing.  

Author Response

Reviewer 1

The work submitted by Kaida Cai et al. entitled A novel LSTM-based machine learning model for the activity prediction of food protein-derived antihypertensive peptides described the work with an LSTM and predicted potential peptide binders for ACE from the source of food. The result would be more impressive if authors could have developed a peptide inhibitor from a short de novo peptide library. Nonetheless, the LSTM model did not resemble experimental data well. Table 2 shows more than five potent peptides that do not agree with experimental data (ratio IC50 >2). Most importantly, the best binder found via experimental study is predicted as a weak binder through the LSTM model (entry 15 in Table 2). I don’t think this model is supreme for predicting a good peptide binder.

Response: Thank you for the critical comments. It has to be admitted that the experimental IC50 value of some randomly synthesized peptides is larger than the predicted value. However, as shown in this study, the mean accuracy, average sensitivity and average specificity of the model was 85.20%, 84.92%, and 85.43%, respectively. Thus, the overall performance of this model is acceptable, which is also better than the other model of antihypertensive peptides [1, 2]. In addition, we think that the limited data we were able to get might impact the accuracy of the model. Hereby, we are searching for more data to improve the performance of the model.

Few comments:

In many places, the IC50 values range has been given, but the unit is missed.

Response: The missing units have been added.

The images are blurry.

Response: The resolution of the images have been improved.

Authors would have considered the SAR studies to improve the binding ability and IC50 value of a selected best peptide binder.

Response: The long short-term memory (LSTM) algorithm was used in our study. LSTM has the advantage of capturing historical information from prior inputs to influence the current input and outputting applications for speech recognition, natural language processing and time series prediction [3]. Since the sequence of the peptide is a predominant factor for the peptide activity, the relationship between the sequence and the activity of the peptide fits the LSTM algorithm. In addition, the LSTM-based model only requires limited features of the peptide, which could simplify the model.

The computational model needs to be updated and attained more specific algorithms for better prediction of peptide binder and close to experimentally validated value.   

Response: Thank you for the important comment. It has to be admitted that the model constructed in this study has limitations, an underlying reason is that the quantity of the training data is limited. Thus, we are trying to get the access of more data to further improve the specificity of the algorithms in the follow-up studies.

References

  1. Kumar, R.; Chaudhary, K.; Singh Chauhan, J.; Nagpal, G.; Kumar, R.; Sharma, M.; Raghava, G. P., An in silico platform for predicting, screening and designing of antihypertensive peptides. Scientific reports 2015, 5, (1), 1-10.
  2. Manavalan, B.; Basith, S.; Shin, T. H.; Wei, L.; Lee, G., mAHTPred: a sequence-based meta-predictor for improving the prediction of anti-hypertensive peptides using effective feature representation. Bioinformatics 2019, 35, (16), 2757-2765.
  3. Tian, C.; Ma, J.; Zhang, C.; Zhan, P., A deep neural network model for short-term load forecast based on long short-term memory network and convolutional neural network. Energies 2018, 11, (12), 3493.

Reviewer 2 Report

Antihypertensive peptides are important bioactive peptides. In this study, the authors applied a LSTM machine learning model for IC50 prediction. The model performance was evaluated in the test set as well as randomly synthesized peptides. 

- How are the 2327 peptides being split into training and testing? In line 99, it seems that the test set consists of 343 peptides. It’s weird to have 343/2327=14.7%.

- In Table 1 and 2, I don’t think it make sense to take predicted IC50 / reported IC50 as a metric. Why did the authors not use RMSE, and possibly compare table 1 and 2 values with their 5-fold CV training score to see if there’s overfitting?

- Based on the above point, then why did the authors calculate accuracy, sensitivity, and specificity in line 97-98. Please use consistent metrics in the whole study. 

- It’s not clear if this is a binary classification or regression task. For sensitivity and specificity metrics, the task should be classification. For the ratio metric, the task should be regression. 

- Figure 2 does not include abundant information. The authors should move this to supporting information. 

- Please keep consistent format, either IC50 or IC_50 (50 as subscript). 

- Please check the writing, such as line 146: As, not AS. 

Author Response

Reviewer2

Antihypertensive peptides are important bioactive peptides. In this study, the authors applied a LSTM machine learning model for IC50 prediction. The model performance was evaluated in the test set as well as randomly synthesized peptides.

- How are the 2327 peptides being split into training and testing? In line 99, it seems that the test set consists of 343 peptides. It’s weird to have 343/2327=14.7%.

Response: Totally, 3429 peptides were retrieved from the database (described in Line 76). As we described in the methodology part, the data was randomly divided into the training set and the validation set of the model in a ratio of 9:1. Thus, 343 peptides were included in the test set. Although the IC50 value of 2327 peptides is less than 100 μM, which is considered as functional ACE inhibitory peptides, the whole benchmark dataset (with 3429 peptides) was used for the study.

- In Table 1 and 2, I don’t think it make sense to take predicted IC50 / reported IC50 as a metric. Why did the authors not use RMSE, and possibly compare table 1 and 2 values with their 5-fold CV training score to see if there’s overfitting?

Response: Thank you for the important comment. The ratio of the predicted IC50 and the reported IC50 was shown to indicated the accuracy of the model in assessing the activity of the specific peptide. As suggested by the reviewer, we added RMSE to evaluate the model performance (The result and method were described in In line 96 and 256, respectively.).

- Based on the above point, then why did the authors calculate accuracy, sensitivity, and specificity in line 97-98. Please use consistent metrics in the whole study.

Response: The accuracy, sensitivity, specificity and RMSE were used to evaluate the performance of the model in general. While, the ratio of the predicted IC50 and the reported IC50 was used to show the accuracy with regard to the specific peptide. The model was evaluated in two dimensions discussed above.

- It’s not clear if this is a binary classification or regression task. For sensitivity and specificity metrics, the task should be classification. For the ratio metric, the task should be regression.

Response: The prediction was defined as “accurate” when the ratio matric was within the range of 0.75 and 1.25, otherwise it is treated as "inaccurate". In this way, the regression task, was converted to the classification task, which was further used for the 5-fold cross validation. The above descriptions have been added in lines 248-251.

- Figure 2 does not include abundant information. The authors should move this to supporting information.

Response: Figure 2 shows the changes of train loss and test loss with the increase of training cycle, which could demonstrate the training efficiency. Thus, we prefer to keep this figure in the main text.

- Please keep consistent format, either IC50 or IC_50 (50 as subscript).

Response: The format of IC50 has been corrected for consistence.

- Please check the writing, such as line 146: As, not AS.

Response: The typo has been corrected.

Round 2

Reviewer 1 Report

The authors have now improved the article as suggested, and the work may be accepted for publication.

Congratulations! to the authors.

Nothing to comment on.

Reviewer 2 Report

The authors have addressed my points.